# Biofilms deform soft surfaces and disrupt epithelia

Alice Cont, Tamara Rossy, Zainebe Al-Mayyah, Alexandre Persat*

Institute of Bioengineering and Global Health Institute, School of Life Sciences, Ecole Polytechnique Fédérale de Lausanne, Lausanne, Switzerland

**Abstract** During chronic infections and in microbiota, bacteria predominantly colonize their hosts as multicellular structures called biofilms. A common assumption is that biofilms exclusively interact with their hosts biochemically. However, the contributions of mechanics, while being central to the process of biofilm formation, have been overlooked as a factor influencing host physiology. Specifically, how biofilms form on soft, tissue-like materials remains unknown. Here, we show that biofilms of the pathogens *Vibrio cholerae* and *Pseudomonas aeruginosa* can induce large deformations of soft synthetic hydrogels. Biofilms buildup internal mechanical stress as single cells grow within the elastic matrix. By combining mechanical measurements and mutations in matrix components, we found that biofilms deform by buckling, and that adhesion transmits these forces to their substrates. Finally, we demonstrate that *V. cholerae* biofilms can generate sufficient mechanical stress to deform and even disrupt soft epithelial cell monolayers, suggesting a mechanical mode of infection.

## Introduction

In their natural environments, bacteria commonly grow and self-organize into multicellular structures called biofilms (*Flemming et al., 2016*). Biofilms form when bacteria attach onto a solid surface and divide while embedding themselves in a matrix of extracellular polymeric substances (EPS) (*O'Toole et al., 2000*). The biofilm matrix is a viscoelastic material generally composed of a mixture of polysaccharides, proteins, nucleic acids and cellular debris (*Flemming and Wingender, 2010*). EPS maintains cell-cell cohesion throughout the lifecycle of a biofilm, also making the resident cells more resilient to selective pressures. The biofilm lifestyle provides resident cells with fitness advantages compared to their planktonic counterpart, for example by increasing their tolerance to external chemical stressors such as antimicrobials and host immune effectors. In addition, its mechanical strength and cohesion promotes biofilm integrity against physical challenge such as flow and grazing (*Mah and O'Toole, 2001*).

Bacteria often colonize the tissues of their host in the form of biofilms. For example, they are a common contributor of infections, as in cystic fibrosis patients who are chronically infected by biofilms of the opportunistic pathogen *P. aeruginosa* (*Donlan and Costerton, 2002*; *Bjarnsholt et al., 2013*). Biofilms are also widespread in microbiota, for example as commensals seek to stably associate with host intestinal epithelium (*De Weirdt and Van de Wiele, 2015*). As they grow on or within a host, biofilms must cope with a battery of chemical and physical stressors. In particular, they must inevitably form at the surface of soft biological material composed of host cells or extracellular matrix (ECM). Multiple biophysical explorations have demonstrated the importance of biofilm internal mechanics in morphogenesis (*Yan et al., 2019*; *Asally et al., 2012*; *Douarche et al., 2015*; *Yan et al., 2016*). However, and despite host-associated biofilms ubiquitously forming on soft surface, we still lack a rigorous understanding of how the mechanical properties of a substrate impacts the physiology of a biofilm, and reciprocally how biofilms impact the mechanics of soft biological surfaces.

*For correspondence:
alexandre.persat@epfl.ch

**Competing interests:** The authors declare that no competing interests exist.

The growth of single cells embedded within self-secreted EPS drives biofilm formation. During this process, cells locally stretch or compress the elastic matrix, thereby exerting mechanical stress (*Douarche et al., 2015*; *Rivera-Yoshida et al., 2018*). This local action at the level of single cells collectively generates mechanical stress across the whole biofilm structure. Thus, the combination of biofilm growth and matrix elastic properties imposes buildup of internal mechanical stress (*Dufrêne and Persat, 2020*). As a consequence of this stress, bacterial colony biofilms form folds and wrinkles when growing on agar plates or at an air-liquid interface (*Yan et al., 2019*; *Trejo et al., 2013*). These mechanics also influences the spatial organization of single cells within *V. cholerae* immersed biofilms (*Hartmann et al., 2019*; *Beroz et al., 2018*; *Drescher et al., 2016*). Internal mechanical stress can also arise by a combination of cell-surface adhesion and growth, influencing the architecture of submerged biofilms and microcolonies. Friction force between the microcolony and the surface opposes biofilm expansion, generating an inward internal stress that leads to a buckling instability verticalizing or reorienting contiguous cells (*Beroz et al., 2018*; *Duvernoy et al., 2018*). These studies demonstrate the importance of mechanics in biofilm morphogenesis and spatial organization, but their function in the context of host colonization remains unknown.

Here, we investigate how biofilms form at the surface of soft material whose mechanical properties replicate the ones encountered in vivo. We show that biofilms from the model pathogens *V. cholerae* and *P. aeruginosa* can deform soft synthetic hydrogel substrates they grow on. By spatially and quantitatively measuring substrate morphology, we propose a model where biofilms buckle to initiate deformations. By comparing wild-type, EPS matrix hypersecreting and mutant strains, we demonstrate that matrix components maintaining cell-cell cohesion and cell-surface adhesion contribute to the mechanism of substrate deformation. The magnitude of the deformations depends on the stiffness of the material in a range that is physiologically-relevant. Using traction force microscopy, we show that biofilms can generate large mechanical stress reaching up to 100 kPa. Finally, we demonstrate that biofilms can deform and even damage tissue-engineered soft epithelia whose mechanics reproduce the ones of a host tissue. These insights suggest that forces generated by growing biofilms could play a role not only in their morphogenesis, but also in mechanically compromising the physiology of their host.

## Results

### Biofilms deform soft substrates

To understand how biofilms interact with soft surfaces, we first explored their formation on synthetic hydrogel substrates. We generated polyethylene glycol (PEG) hydrogel films via photoinitiated polymerization of PEG diacrylate precursors at the bottom surface of microfluidic channels. These polymeric films are covalently bound to the glass surface to avoid drift and delamination. By using a 'sandwich' method for polymerization, we could fabricate flat ~100-μm-thin PEG films that allowed us to perform high resolution live confocal imaging of biofilm formation under flow (*Figure 1A*). We first used a rugose variant of *V. cholerae* A1152 strain (referred to as *V. cholerae Rg*) which constitutively secretes EPS matrix, thereby forming robust and reproducible biofilms. On soft hydrogels, *V. cholerae Rg* formed biofilms whose bottom surfaces appeared bell-shaped (*Figure 1B*), in striking difference with the typically flat-bottom biofilms that form on hard surfaces such as glass and plastic. To distinguish whether this shape was a result of the deformation of the hydrogel or of the detachment of the biofilm from the surface, we embedded fluorescent tracer particles within the hydrogel film by mixing them with the pre-polymer solution before the cross-linking step. We could observe that the fluorescent tracer particles filled the apparent bell-shaped void at the biofilm core and that the hydrogel surface and the biofilm remained in contact (*Figure 1C*). This demonstrates that the soft hydrogel substrate deforms under *V. cholerae* biofilms.

We then wondered whether these deformations were specifically induced by *V. cholerae* or could represent a common feature of biofilms across species. We thus tested whether *P. aeruginosa* biofilms could deform soft hydrogels. We found that biofilms of *P. aeruginosa wspF*⁻ mutant (*P. aeruginosa Rg*), which constitutively produces EPS matrix and robustly forms biofilms, could similarly deform soft PEG hydrogels (*Figure 1D–E*). In summary, *V. cholerae* and *P. aeruginosa*, two model biofilm-forming species with distinct EPS composition are both able to deform soft substrates. This

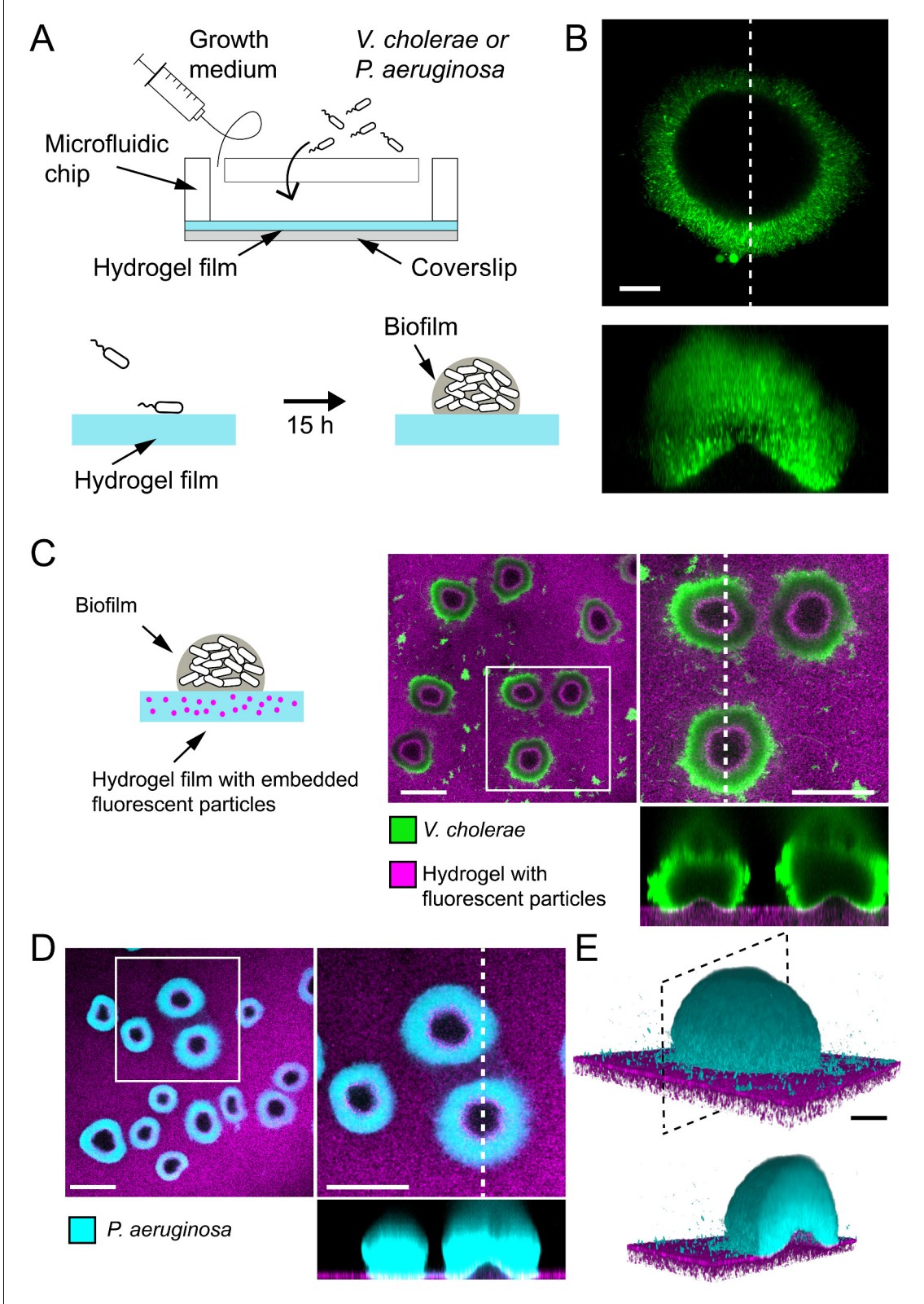

**Figure 1.** Biofilms deform soft substrates. (**A**) Illustration of experimental setup where we generate thin hydrogel films at the bottom surface of microchannels. These devices allow us to study biofilm formation on hydrogels reproducing mechanical properties of host tissues. (**B**) In-plane and cross-sectional confocal visualizations show that *V. cholerae Rg* biofilms growing on hydrogels display large gaps at their core. (**C**) Embedding fluorescence tracer particle in the hydrogel films allow for visualization of deformations. *V. cholerae Rg* biofilms formed at the surface of the films

*Figure 1 continued on next page*

*Figure 1 continued*

deform the substrate. (D) *P. aeruginosa Rg* biofilms similarly deform the soft substrates. Hydrogel elastic modulus: (**B and C**) $E$ = 12 kPa, (**D and E**) $E$ = 38 kPa. Scale bars: (**C and D**) 100 µm, (**B and E**) 20 µm.

is consistent with a mechanism where biofilms generate mechanical stress on the material they grow on.

## Biofilms deform soft substrates after reaching a critical diameter

How do biofilms mechanically deform hydrogel films? Given the influence of growth-induced internal mechanical stress on biofilm morphology and architecture, we hypothesized that biofilms could deform soft substrates by transmission of internal stresses to the substrate they grow on. To test this hypothesis, we performed dynamic visualizations of the deformations of hydrogel films as biofilms grew. To obtain an accurate deformation profile, we performed a radial re-slicing and averaging around the biofilm center. We could thus extract the deformation profile $\delta$, its maximum deformation amplitude $\delta_{max}$ and full-width at half maximum $\lambda$ (*Figure 2A*). We thus recorded surface profiles for many biofilms. By reconstructing hydrogel surfaces for biofilms of different sizes, we found that $\delta_{max}$ and $\lambda$ linearly scaled with the diameter $d$ of the biofilm (*Figure 2—figure supplement 1*), indicating that biofilm expansion promotes surface deformation.

We went further and dynamically tracked these deformations for single biofilms. Deformations increased as biofilms grew, even displaying a slight recess near the biofilm edges (*Figure 2B-C*, *Video 1*). In these visualizations, we noticed that there was a lag between the increase in biofilm diameter and the onset of deformation, with a finite deformation only appearing after 7 hr of growth. This was further confirmed by following the deformations generated by many biofilms. Measurable morphological changes of the surface appeared after 6 to 7 hr of growth (*Figure 2D*). Rescaling these measurements with the diameter of the biofilm collapsed $\delta_{max}$ measurements, highlighting a critical biofilm diameter (35 µm) above which deformations emerged (*Figure 2E*). The existence of a critical diameter is reminiscent to buckling instabilities of rigid bodies subject to compressive stress, as in Euler buckling.

## Biofilms push their substrate in the growth direction

To further investigate the mechanism by which biofilms deform surfaces, we quantified the hydrogel substrate strain during growth. To achieve this, we tracked the displacements of the fluorescent tracer particles embedded within the hydrogel in 3D using a digital volume correlation algorithm (*Toyjanova et al., 2014*). At the early stages of hydrogel deformation, we found that in the plane defined by the initial surface at rest, the particles under the biofilm move in the direction of growth. Thus, the strain field shows that the biofilm stretches its substrate radially in the outward direction in addition to vertical deformations (*Figure 2F* and *Figure 2—figure supplement 2*). In other words, a biofilm applies an in-plane stress on the substrate in its growth direction, which is most likely generated by a friction between the biofilm and the surface (*Beroz et al., 2018*; *Duvernoy et al., 2018*). As a result, the elastic biofilm experiences a force in the opposite direction, toward its center. This result is consistent with observations of growth of colony biofilms (*Yan et al., 2019*). In summary, the opposition between biofilm growth and friction with the surface generates an internal mechanical stress within the biofilm oriented radially, towards its center.

## Wild-type and rugose biofilms deform soft-substrates

Given its importance in generating internal stress with the biofilm, we anticipate that the EPS matrix plays a role in the onset of substrate deformations. Rugose strains constitutively produce copious amounts of matrix compared to wild-type (*Yan et al., 2016*; *Hartmann et al., 2019*; *Teschler et al., 2015*; *Yildiz and Schoolnik, 1999*; *Pestrak et al., 2018*; *Starkey et al., 2009*; *Yan et al., 2019*). To verify whether WT can deform substrates and to probe the contribution of matrix hypersecretion, we quantitatively compared the deformations of WT strains of *V. cholerae* and *P. aeruginosa* with their *Rg* forms. First, WT *P. aeruginosa* deformed hydrogel films under the same growth conditions as *Rg*. The amplitude of the deformation $\delta_{max}$ was substantially lower than *Rg* (*Figure 3A*). Then, we visualized smooth WT *V. cholerae* A1552 under multiple growth conditions which are known to

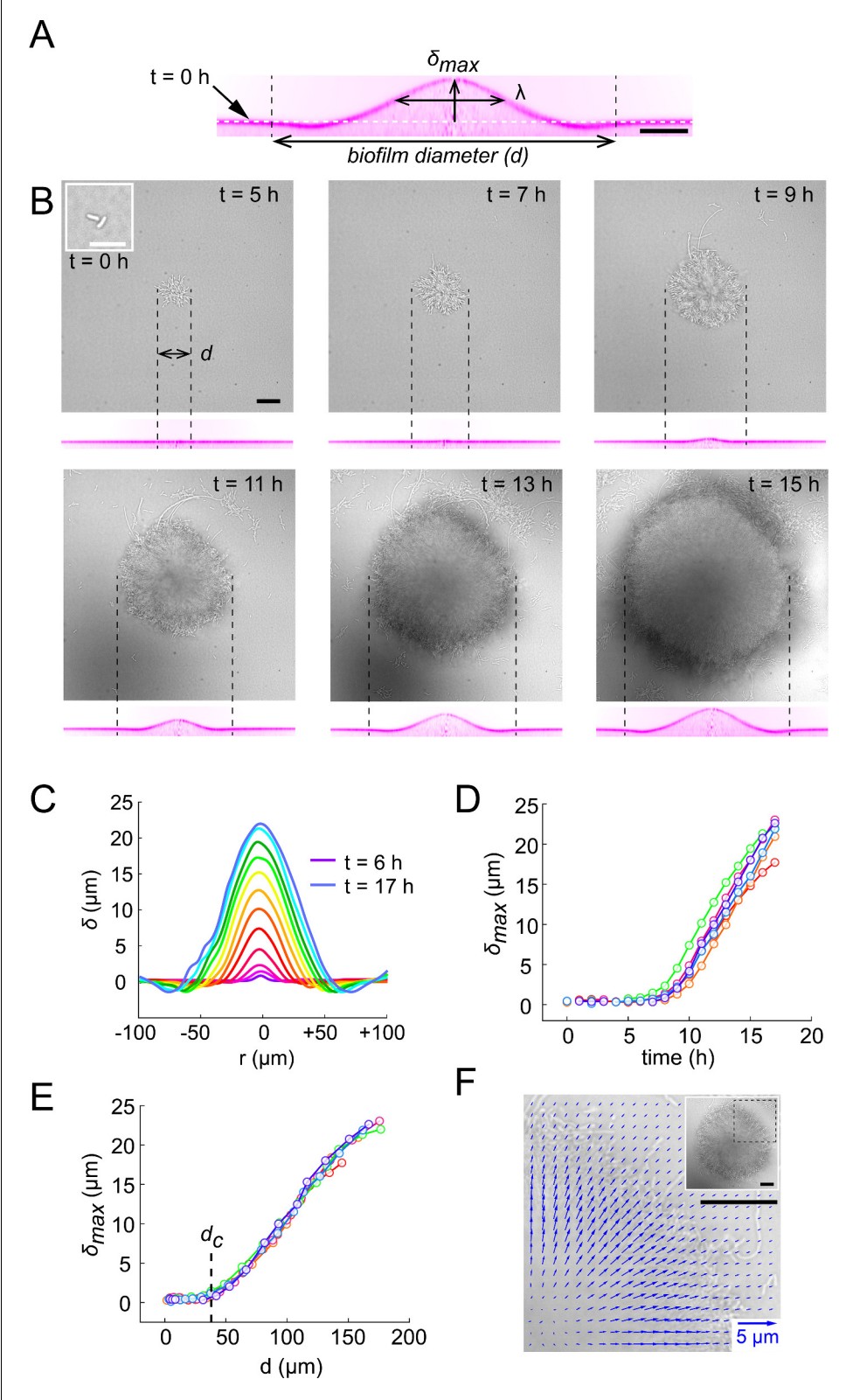

**Figure 2.** Biofilms deform their substrate by buckling. (A) Morphological parameters $\delta_{max}$ (maximum deformation amplitude) and $\lambda$ (half max full width) computed from resliced deformation profiles. Dashed line indicates the baseline position of the gel surface. (B) Timelapse visualization of *V. cholerae Rg* biofilm growth (brightfield, top) with deformation (reslice, bottom). Dashed lines indicate biofilm position and size on the corresponding hydrogel profile. (C) Superimposition of these profiles shows the rapid deformation and the emergence of a recess at biofilm edges. Each color corresponds to

*Figure 2 continued on next page*

*Figure 2 continued*

the same biofilm at different times. (D) Time evolution of $\delta_{max}$ shows a rapid increase after 6 to 7 hr of growth. (E) The dependence of $\delta_{max}$ on biofilm diameter highlights a critical biofilm diameter $d_c$ above which deformation occurs. For D and E, each line color corresponds to a different biofilm. (F) Hydrogel strain field computed by digital volume correlation between 11 hr and 12 hr of growth. We superimposed the vector strain field with a brightfield image of the biofilm. For visualization purposes we only display data for the top right quarter of the biofilm shown in inset (dashed lines). $E$ = 38 kPa. Scale bar: 10 μm for inset $t$ = 0 hr in (B), else 20 μm.

The online version of this article includes the following source data and figure supplement(s) for figure 2:

**Source data 1.** Deformation amplitude and $\lambda$ as a function of time and diameter.
**Figure supplement 1.** Biofilm diameter-dependence of $\delta_{max}$ and $\lambda$.
**Figure supplement 2.** Hydrogel deformation field computed at different growth stages, superimposed with a brightfield image of the biofilm.

influence its ability to form robust biofilms. WT biofilms deformed the hydrogel substrate when grown in M9 medium, but not in LB where biofilms are rare and WT *V. cholerae* essentially spread as a monolayer of cells on the surface (*Figure 3B*). The deformation amplitude $\delta_{max}$ of WT biofilms in M9 could reach the magnitude of $Rg$ (*Figure 3C*), but were however more heterogeneous between biofilms. Finally, we went on to compare the ability of smooth WT *V. cholerae* strains A1552, N16961, and C6706, to deform hydrogel substrates. These strains are commonly used as models for biofilm formation, build biofilms with identical matrix components, but likely regulate them distinctly. We found that all three strains could deform the substrate (*Figure 3—figure supplement 1*). The magnitudes of deformation however differed between isolates: N16961 and C6706 deform the substrate more dramatically than A1552. In summary, WT and rugose *V. cholerae* and *P. aeruginosa* biofilms deform soft substrates. The quantitative differences in deformation amplitudes observed across growth conditions and strains, plus the established role of the EPS in biofilm morphogenesis suggests that matrix production plays a fundamental role in generating internal forces and in subsequently deforming soft substrates.

## EPS composition drives biofilm and substrate deformations

We then wondered how matrix composition and associated changes in mechanical properties of biofilms influence substrate deformations. To investigate their contributions, we used *V. cholerae* EPS matrix mutants with altered biofilm structure and mechanical properties. The *V. cholerae* matrix is mainly composed of a polysaccharide (*vps*) and proteins including Rbma, an extracellular component which specifically strengthens cell-cell cohesion and stiffens the matrix (*Teschler et al., 2015*; *Yan et al., 2018*). We confirmed that *vpsL* deletion mutants couldn't form biofilms (*Figure 4—figure supplement 1A*). More surprisingly, we found that biofilms of *rbmA* deletion mutants were unable to deform the hydrogel substrate, indicating that cell-cell cohesion is an essential ingredient in force generation (*Figure 4A*). Complementation of *rbmA* restored the ability to deform (*Figure 4—figure supplement 1B*). In *P. aeruginosa*, the polysaccharides Pel and Psl, and the protein CdrA play partially redundant functions in maintaining elastic properties of the biofilm (*Jennings et al., 2015*; *Kovach et al., 2017*; *Colvin et al., 2012*). In a similar manner, we found that the deformations generated by *P. aeruginosa pel*, *psl* and *cdrA* mutants were lower compared to $Rg$, but were not abolished (*Figure 4B–C*). The largest drop in deformation occurred in the *pel* mutant. This further demonstrates that mechanical cohesion provided by the EPS matrix plays a key role in surface deformation.

We then probed the function of adhesion of the biofilm with the surface by visualizing the deformations generated by a *V. cholerae bap1* deletion mutant. Bap1 is specifically secreted at the biofilm-substrate interface to maintain proper surface attachment (*Teschler et al., 2015*). The *bap1* mutant formed biofilms that did not deform the surface. However, it produced slightly bent biofilms delaminated from

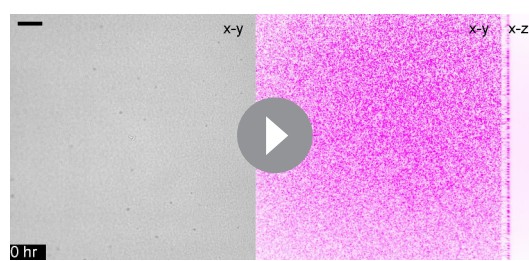

**Video 1.** Timelapse visualization of *V. cholerae Rg* biofilm growth (brightfield) and corresponding hydrogel deformation ($E$ = 38 kPa) in the *xy* and *xz* planes. Scale bar 20 μm.
https://elifesciences.org/articles/56533#video1

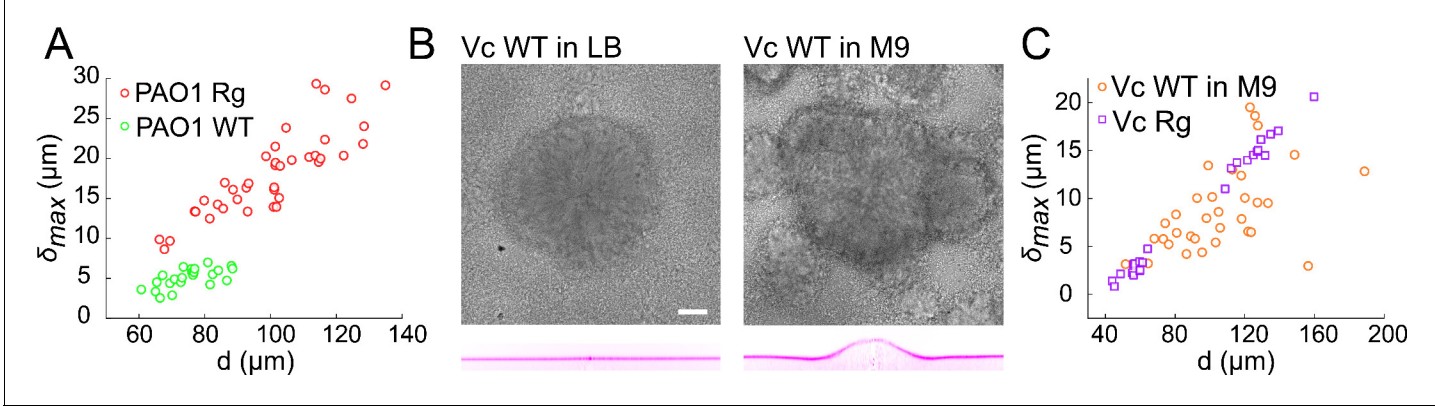

**Figure 3.** Wild-type and rugose biofilms deform soft-substrates. (A) Biofilm diameter-dependence of maximum deformation for rugose and smooth variants of *P. aeruginosa*. (B) Smooth variant of *V. cholerae* A1552 deforms hydrogels when growing in M9 medium, but not in LB. (C) Biofilm diameter-dependence of maximum deformation for rugose and smooth variants of *V. cholerae*. Data points correspond to biofilms grown in two microfluidic chambers for PAO1 Rg, PAO1 WT and Vc WT and to biofilms grown in one microfluidic chamber for Vc Rg. E = 38 kPa. Scale bars: 20 μm.
The online version of this article includes the following source data and figure supplement(s) for figure 3:

**Source data 1.** Deformation amplitude for WT strains.
**Figure supplement 1.** Biofilm diameter-dependence of maximum deformation for the smooth variant of different *V. cholerae* strains grown in M9.

the substrate creating a gap between the biofilm and the hydrogel, indicating that it may have buckled (*Figure 4A*). Complementation of *bap1* restored the ability to deform the hydrogel (*Figure 4—figure supplement 1C*). Our observations of delamination of the *bap1* mutant show that adhesion transmits mechanical stress generated by buckling from the biofilm to the substrate. Due to the redundant functions of its EPS components, we could not produce *P. aeruginosa* mutants with altered surface adhesion properties. However, *P. aeruginosa* biofilms growing on hydrogels with large Young's modulus delaminated (*Figure 4—figure supplement 2*). This observation highlights that the transition between deformation and delamination depends on the relative contribution of adhesion strength and substrate elasticity. In summary, cell-cell mechanical cohesion is essential in generating the internal stress that promotes biofilm buckling, while cell-substrate adhesion transmits this stress to the underlying substrate (*Figure 4D*).

## Biofilms generate large traction forces

Biofilms thus deform soft materials by coupling growth-induced buckling and adhesion to their substrate. Could the mechanical stress generated on the substrate also impact various types of biological surfaces? To first explore this possibility, we quantified the forces exerted by the biofilm on hydrogel films. We used our particle tracking data to perform traction force microscopy (TFM), thereby computing the stress field and surface forces applied by the biofilm on the hydrogel. Traction forces were surprisingly large, reaching 100 kPa at the biofilm center after 12 hr of growth (*Figure 5A*). We note that the magnitude of the stress is relatively large, reaching a value close to the typical turgor pressure which in essence drives biofilms growth and stretching (*Rojas and Huang, 2018*). In comparison, epithelial cell-cell junctions break when experiencing a few kPa (*Charras and Yap, 2018*). Therefore, we anticipate that biofilms produce sufficient force to mechanically deform and potentially dismantle epithelia.

Given the large forces generated by biofilms on hydrogel substrates, we wondered to which extent they could deform biomaterials of different stiffnesses as defined by their Young's modulus. To test this, we reproduced the mechanical properties of various tissue types by tuning the stiffness of the PEG hydrogel films between E = 10 kPa and E = 200 kPa (*Discher et al., 2005*; *Lee et al., 2014*). The stiffest hydrogels only slightly deformed (*Figure 5B*, $\delta_{max}$ = 5 μm for E = 203 kPa). In contrast, biofilms growing on the softest hydrogels displayed large deformations ($\delta_{max}$ = 27 μm for E = 12 kPa). The rate of increase of deformations was inversely correlated with stiffness, resulting in differences in $\delta_{max}$ between colonies of identical diameter growing on substrates with distinct stiffnesses (*Figure 5C*). For each stiffness, the deformation amplitude $\delta_{max}$ and the width $\lambda$ increased

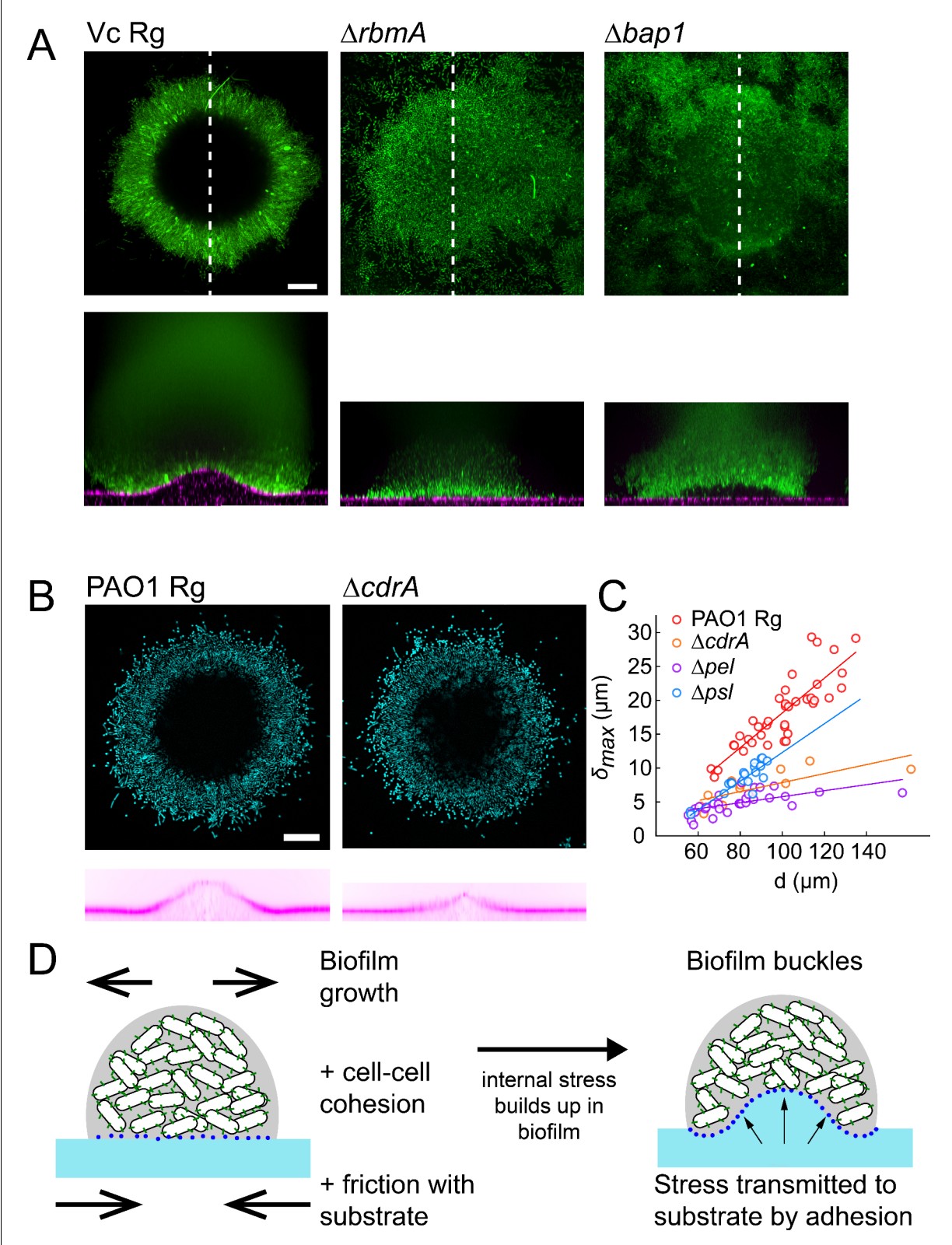

**Figure 4.** EPS composition drives biofilm and substrate deformations. (A) Deformations of hydrogel substrates by *V. cholerae Rg*, *rbma⁻* and *bap1⁻* biofilms. Biofilms formed by *rbma⁻* and *bap1⁻* fail to deform the substrate. *bap1⁻* biofilms delaminate from the hydrogel surface. (B) Comparison of hydrogel deformations by *P. aeruginosa Rg* and *cdrA⁻* biofilms. (C) Dependence of maximum deformations on *P. aeruginosa Rg*, *cdrA⁻*, *pel⁻* and *psl⁻* biofilm diameter. All matrix mutants tend to generate weaker deformations compared to *Rg*. Data points correspond to different biofilms grown in two

*Figure 4 continued on next page*

*Figure 4 continued*

microfluidic chambers. (**D**) A model for the mechanism of biofilm deformation of soft substrates. Buildup of mechanical stress in the biofilm induces buckling. Adhesion between the biofilm and the surface transmits buckling-generated stress to the hydrogel, inducing deformations. $E = 38$ kPa. Scale bars: 20 μm.

The online version of this article includes the following source data and figure supplement(s) for figure 4:

**Source data 1.** Deformation amplitude for PAO1 matrix mutants.
**Figure supplement 1.** Deformation behaviour for *vpsL* deletion mutant and complementation strains.
**Figure supplement 2.** *P. aeruginosa* biofilms on substrates with different stiffness.

linearly with biofilm diameter (*Figure 5C* and *Figure 5—figure supplement 1*). Rescaling $\delta_{max}$ with the biofilm diameter highlights a power-law relationship between deformation and substrate stiffness (*Figure 5—figure supplement 2*).

## Biofilms deform and disrupt epithelial cell monolayers

Given the ability of biofilms to generate large forces and to deform materials across a wide stiffness range, we wondered whether they could disrupt soft epithelium-like tissues. To test how biofilms can mechanically perturb host tissue during colonization, we engineered epithelial cell monolayers at the surface of a soft extracellular matrix (ECM). This system replicates the mechanical properties of host epithelia including tissue stiffness and adhesion to underlying ECM. As a result, it constitutes a more realistic host-like environment compared to cell monolayers grown on plastic or glass. We thus engineered epithelial monolayers of enterocyte-like CMT-93 cells on a soft extracellular matrix composed of Matrigel and collagen (*Figure 6A*). This produced soft and tight ECM-adherent epithelia. We seeded the surface of these epithelia with *V. cholerae Rg*. We note that the *Rg* strain has reduced virulence compared to WT *V. cholerae* due to its constitutively high levels of cyclic-di-GMP which decrease the expression of virulence factors (*Tischler and Camilli, 2005*). *V. cholerae* biofilms formed at the epithelial surface within 20 hr (*Figure 6B*). Overall, biofilms perturbed the shape of the epithelium. Under biofilms, the cell monolayer detached from its ECM substrate and was often bent as did synthetic hydrogel films (*Figure 6B–ii*). More surprisingly, we observed that CMT-93 cell

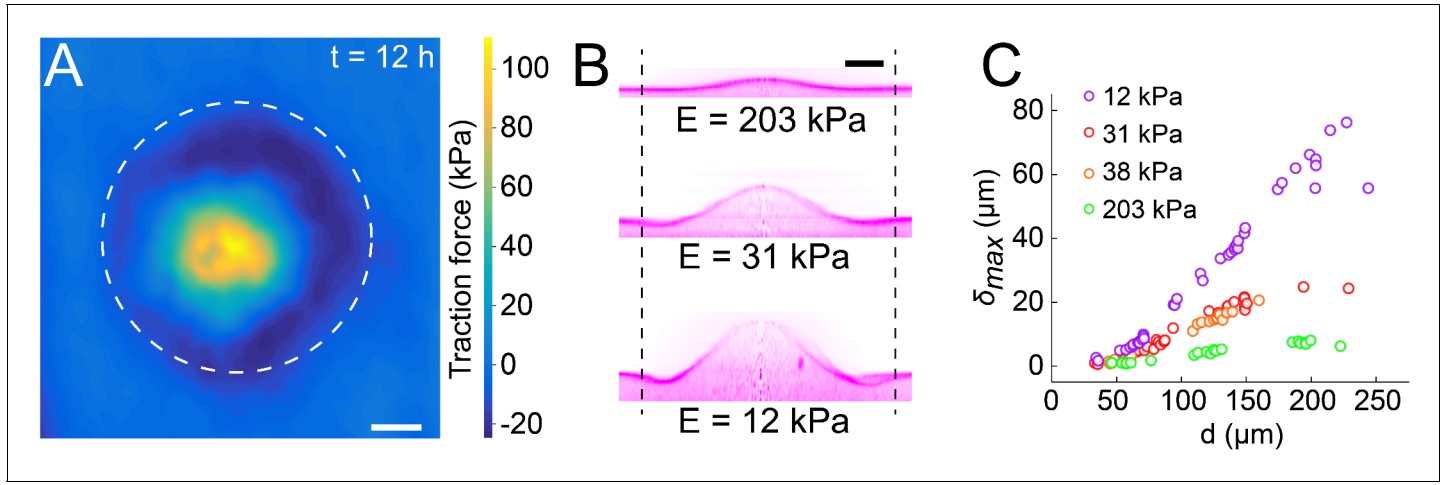

**Figure 5.** Biofilms generate large traction forces. (**A**) Traction force microscopy measurements at the hydrogel-biofilm interface. The dashed line shows the edge of the biofilm. Traction force is largest at the biofilm center, reaching 100 kPa. (**B**) Deformation profiles generated by *V. cholerae Rg* biofilms of equal diameters on three hydrogels with different stiffness. (**C**) Biofilm diameter-dependence of maximum deformation for four different hydrogel composition representing a typical range of tissue stiffnesses. The softest hydrogel can deform up to 80 μm for a biofilm diameter of 220 μm. Data points correspond to different biofilms grown in one microfluidic chamber. Scale bar: 20 μm.

The online version of this article includes the following source data and figure supplement(s) for figure 5:

**Source data 1.** Deformation amplitude and $\lambda$ for substrates of different stiffness.
**Figure supplement 1.** Biofilm diameter-dependence of $\lambda$ for substrates with different moduli.
**Figure supplement 2.** Power-law relationship between deformation $\delta_{max}$ and substrate moduli (E).

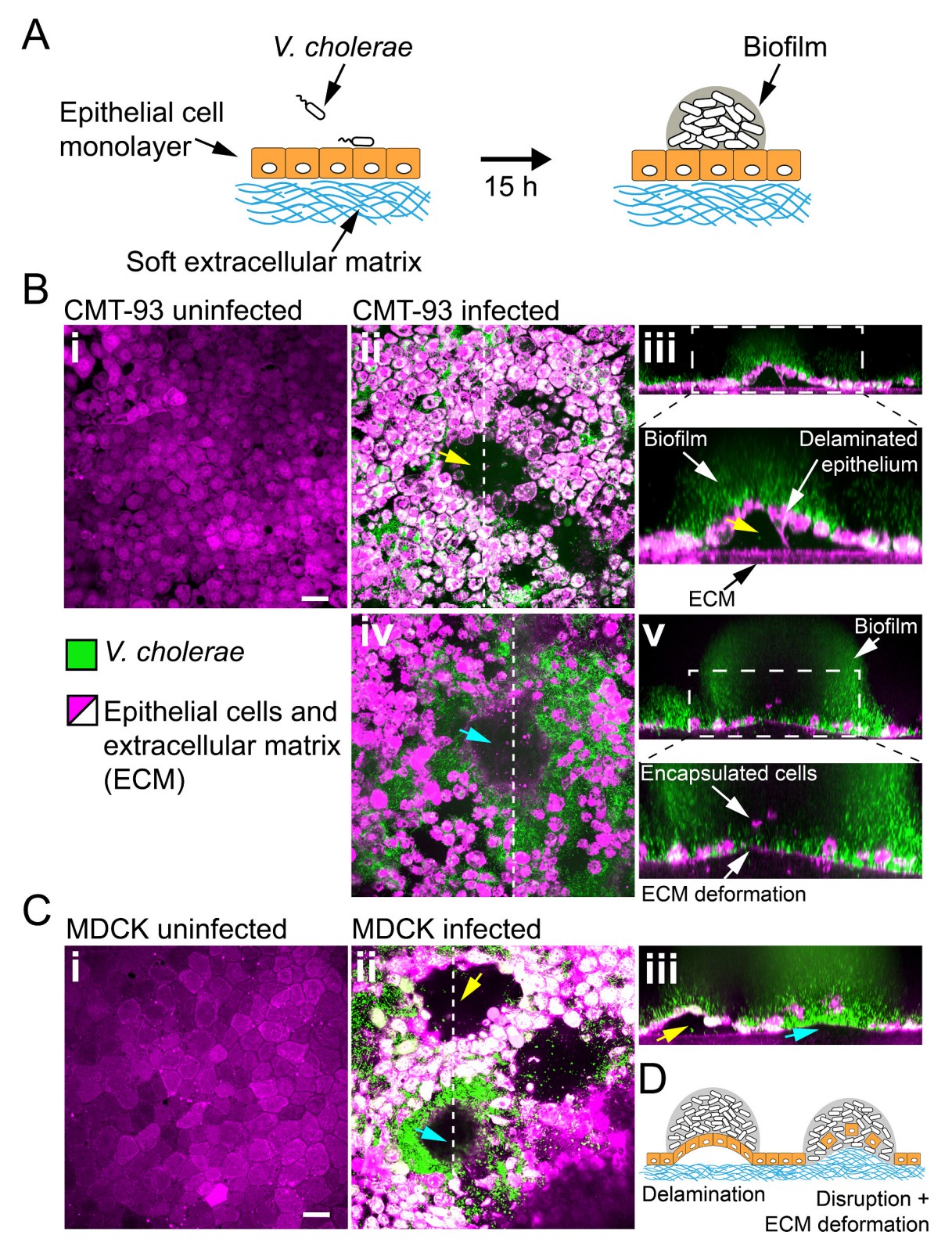

**Figure 6.** Biofilms deform and disrupt epithelial cell monolayer. (**A**) CMT-93 and MDCK cells grow at the surface of a soft ECM into a tight monolayer on which we seed a liquid inoculum of *V. cholerae Rg*. (**B**) Confocal images of uninfected (i) and infected (ii-v) monolayers of CMT-93 cells. Yellow arrow indicates gaps in the epithelial monolayer (ii and iii), blue arrow shows deformed ECM (iv). (**C**) Confocal images of uninfected (i) and infected (ii-iii) monolayers of MDCK cells, also showing delamination and rupture as illustrated in (**D**). Scale bars: 20 μm.

*Figure 6 continued on next page*

*Figure 6 continued*

The online version of this article includes the following figure supplement(s) for figure 6:

**Figure supplement 1.** Confocal images of uninfected (i) and infected (ii-iv) monolayers of Caco-2 cells.
**Figure supplement 2.** Biofilms perturb the viability of MDCK cell monalyers.
**Figure supplement 3.** Biofilms increase the permeability of MDCK cell monolayers.

monolayers lost cohesion and single cells were engulfed by the biofilm. This allowed the biofilm to breach the epithelium and reach the ECM. There, biofilms deformed the ECM substrate, turning the initially flat surface into a dome-like shape as our synthetic hydrogels did (*Figure 6B-iv*). These disruptions did not depend on cell type and species as *V. cholerae* could also damage and bend monolayers of MDCK cells which have strong cell-cell junctions (*Figure 6C*; *Harris et al., 2012*) and human intestinal epithelial cells Caco-2 (*Figure 6—figure supplement 1*).

How do these perturbations compromise the barrier function of the epithelium? We further characterized the integrity of cell monolayers near biofilms by measuring permeability and viability. We assessed cell viability using Calcein-AM which only generates a fluorescent signal in live cells. We essentially observed that cells engulfed in the biofilms were dead (*Figure 6—figure supplement 2*). In contrast, the viability of the cells around the biofilm base was not compromised. We then measured epithelial permeability using FITC – Dextran probes. Dextran could diffuse across infected monolayers into the ECM, in contrast with uninfected conditions (*Figure 6—figure supplement 3i-ii*). In infected monolayers, we could detect fluorescence in the epithelial opening created by the biofilm and between cells as a result of damaged epithelial cell-cell junctions in the vicinity of the biofilm (*Figure 6—figure supplement 3 iii-v*). Our observations suggest that biofilms apply mechanical forces on host tissue which perturb the morphology, integrity and viability of epithelia, as well as its underlying ECM.

## Discussion

We demonstrated that biofilms can deform the surface of soft materials they grow on. We observed that both *V. cholerae* and *P. aeruginosa* generate these deformations, suggesting that it is a feature of biofilm growth and is not species-dependent. We identified key physical and biological components that enable these deformations. In particular, our measurements of hydrogel deformations provide evidence consistent with a mechanism where the biofilm buckles as it grows. This mechanism is reminiscent of Euler buckling where the internal compressive stress in a beam triggers an instability that induces transverse deformations. In our case, we found that the onset of the buckling instability depends on growth under mechanical constraint which generates a buildup of compressive stress. The EPS matrix is a major contributor of the buildup of internal stresses. Indeed, when we compared the rugose variants (matrix overproducer) with the smooth WT strains of *V. cholerae*, we observed a decrease or a complete loss of deformation when grown in LB. In fact, *V. cholerae* does not robustly form biofilms in LB, likely due to poor matrix secretion, making it an unpopular growth medium for flow cell biofilms. In contrast, WT *V. cholerae* biofilms growing in M9 generated deformations comparable to the ones caused by the rugose variant, suggesting an important link with matrix production. The larger variability in deformation amplitudes in WT compared to *Rg* likely originates from heterogeneity in biofilm matrix secretion and regulation.

In-plane hydrogel strain measurements indicate a friction between the surface and the expanding biofilm, which promotes buildup of internal stress. Also, the fact that biofilms of the *V. cholerae rbmA* and *P. aeruginosa* EPS genes deletion mutants have reduced or abolished ability to buckle or to deform the surface indicates that cell-cell cohesion in the biofilm may also participate in mechanical constraint. Without cell-cell cohesion and matrix elastic property, the viscous biofilm would flow, dissipating mechanical stress and eluding the elastic instability.

These two contributions, biofilm-surface friction and matrix elasticity, induce a buildup of compressive stress within the biofilm, ultimately causing buckling. Biofilm adhesion transmits buckling-induced stresses to the substrate, generating deformations. While a full description of the physical deformation mechanism is a complex problem involving biofilm and substrate mechanics, we can already highlight important features from our observations. First, the facts that the onset of

deformation occurs at a finite critical biofilm diameter and that the width of the deformation $\lambda$ scales linearly with this diameter are consistent with an Euler-type buckling instability (*Timoshenko and Gere, 2019*). Also, the slight negative deformations (recess) observed near the edge of larger biofilms is reminiscent of higher order buckling modes. Then, the power-law relationship between deformation and substrate stiffness is qualitatively consistent with the theory of buckling of plates coupled to an elastic foundation (*Wang, 2005*). The delamination of *V. cholerae* mutants and *P. aeruginosa* on stiff substrates highlight the importance of the balance between adhesion and substrate elasticity in force transmission (*Wang and Zhao, 2015*). At the time of delamination, elastic deformation and adhesion energies balance. Thus, biofilm delamination could help estimate the adhesive strengths of specific matrix components. Overall, biofilms mechanically shape their environment via a buckling-adhesion mechanism, reminiscent of the buckling and wrinkling of plates and films on elastic foundations (*Wang, 2005*). A comprehensive understanding of force generation in biofilms and of substrate response will require further development of mechanical models along with measurements of biofilm internal stresses.

Internal stress generated by bacterial expansion under physical constrains influences the morphologies of colony biofilms, forming wrinkles, folds, and blisters. The shapes of these colonies are also caused by a buckling/wrinkling-like instability which depends on the mechanical properties of the matrix. These mechanically generated shapes have been observed in *V. cholerae*, *P. aeruginosa*, *Bacillus subtilis*, and *E. coli* and have been instrumental as an obvious phenotype to identify components and regulators of the biofilm matrix and to characterize the mechanics driving multicellular growth (*Yildiz and Schoolnik, 1999*; *Starkey et al., 2009*; *Serra et al., 2015*; *Wilking et al., 2013*; *Gloag et al., 2018*). However, the impact of these macroscale morphological changes and internal mechanics on the physiology of resident microbes have yet to be identified. Immersed, micrometer scale biofilms that are commonly found in natural microbial niches also undergo architectural transitions due to the emergence of internal mechanical stress. Cell-cell cohesion and cell-substrate adhesion can favor the alignment of single cells within the multicellular structure during biofilm growth. For example, a buckling instability causes *V. cholerae* cell verticalization in the initial step of biofilm formation, in a mechanism that depends on friction of single cells with their glass substrate, generating compressive mechanical stress (*Beroz et al., 2018*). Single cells in *E. coli* microcolonies reorient through a similar mechanism (*Duvernoy et al., 2018*). The physiological functions of these cellular rearrangements have however not yet been identified. The buckling-adhesion model we here propose is consistent with the mechanics of immersed and colony biofilms. Our observations suggest that internal mechanical stress can have a function in the interaction between the biofilm and its surrounding environment, influence the morphology and mechanics of its material substrate. This may result in fouling of abiotic surfaces, in damaging competing biofilms or even host tissues.

Despite being widespread in the environments of microbes, the influence of substrate rigidity is often overlooked in studies of surface attachment and biofilm formation (*Lichter et al., 2008*; *Kolewe et al., 2015*; *Song and Ren, 2014*). Using a materials approach aimed at reproducing a host-like environment, we found that substrate mechanical properties have a strong impact on biofilm development. Biofilm-induced deformations are particularly relevant when considering their growth at the surface of soft biological tissues. We demonstrated that biofilms generate large forces, and that these forces can be transmitted to underlying epithelia. In response, we observed that epithelial monolayers delaminate from their ECM and subsequently bend. The biofilm-generated forces also disrupt epithelial monolayers. Consistent with this, traction force microscopy measurements show that biofilms can generate 100 kPa surface stress, which is larger than the strength of epithelial cell-cell junctions that typically rupture under the kPa stress range (*Harris et al., 2012*). Thus, the biofilm opened cell-cell junctions increasing epithelial permeability. Biofilms caused epithelial cell death when removed from the monolayer. These observations suggest that mechanical forces generated by biofilms can impact epithelial integrity, in addition to other well-known chemical factors such as toxins (*Bischoff et al., 2014*). We note that a more realistic epithelium would be protected by a layer of mucus of very low elasticity, which could also be strongly deformed by biofilms. Pathogen like *V. cholerae* can however swim through the mucus layer to reach the epithelium surface (*Bartlett et al., 2017*).

In summary, our visualizations in tissue-engineered epithelia and on hydrogel films suggest that biofilms could mechanically damage host tissues when growing in vivo. Consistent with this hypothesis, biofilms can cause tissue lesions. For example, the urine of vaginosis patients contains

desquamated epithelial cells covered with biofilms (*Swidsinski et al., 2013*; *Hardy et al., 2016*). Commensal biofilms form scabs at the epithelial surface of honeybee's gut, triggering immune responses (*Engel et al., 2015*). Epithelial integrity is also compromised in intestinal diseases such as inflammatory bowel disease in a process that highly depends on the composition of the microbiota (*Bischoff et al., 2014*; *Chelakkot et al., 2018*; *Ahmad et al., 2017*; *Tremaroli and Bäckhed, 2012*). Finally, hyper-biofilm forming clinical variants of *P. aeruginosa* cause significant damage to the surrounding host tissue despite its reduced virulence (*Pestrak et al., 2018*).

Most studies of biofilm formation have so far focused on their internal organization and mechanics, and on the genetic regulation of matrix production. How biofilms physically interact with their natural environments has been however vastly unexplored, but could vastly contribute to a holistic understanding of host-microbe interactions. Mechanical interactions between bacterial collectives and their host may thus represent an overlooked contributor of infections, colonization, and dysbiosis. Non-pathogenic biofilm-forming species, including commensals, could thus very well induce epithelial damage and in fact contribute to chronic inflammation.

# Materials and methods

## Key resources table

| Reagent type (species) or resource | Designation | Source or reference | Identifiers | Additional information |
|---|---|---|---|---|
| Strain, strain background (*V. cholerae A1552*) | Vc Rg | *Yildiz and Schoolnik, 1999* | | Rugose variant |
| Strain, strain background (*V. cholerae A1552*) | Vc WT | *Yildiz and Schoolnik, 1999* | | Smooth wild-type variant |
| Strain, strain background (*V. cholerae A1552*) | Vc Rg Δ*vpsL* | This study | | In frame deletion of *vpsL* in rugose background obtained by matings of Vc Rg with S17 harboring deletion plasmid pFY_922 |
| Strain, strain background (*V. cholerae A1552*) | Vc Rg Δ*rbmA* | This study | | In frame deletion of *rbmA* in rugose background obtained by matings of Vc Rg with S17 harboring deletion plasmid pFY_113 |
| Strain, strain background (*V. cholerae A1552*) | Vc Rg Δ*bap1* | This study | | In frame deletion of *bap1* in rugose background obtained by matings of Vc Rg with S17 harboring deletion plasmid pFY_330 |
| Strain, strain background (*V. cholerae A1552*) | Vc Rg Δ*rbmA* pBAD*rbmA* | This study | | Vc Rg Δ*rbmA* harboring the plasmid pNUT1236 |
| Strain, strain background (*V. cholerae A1552*) | Vc Rg Δ*bap1* pBAD*bap1* | This study | | Vc Rg Δ*bap1* harboring the plasmid pBAD/Myc-His B |
| Strain, strain background (*V. cholerae N16961*) | N16961 | *Drescher et al., 2016* | | Smooth wild-type variant |
| Strain, strain background (*V. cholerae C6706*) | C6706 | *Thelin and Taylor, 1996* | | Smooth wild-type variant |
| Strain, strain background (*P. aeruginosa*) | PAO1 WT | *Hickman et al., 2005* | | |
| Strain, strain background (*P. aeruginosa*) | PAO1 Rg | *Rybtke et al., 2012* | | In frame deletions of *wspF* |

*Continued on next page*

*Continued*

| Reagent type (species) or resource | Designation | Source or reference | Identifiers | Additional information |
|---|---|---|---|---|
| Strain, strain background (*P. aeruginosa*) | PAO1 Rg Δ*pel* | *Rybtke et al., 2012* | | In frame deletions of *wspF*, *pelA* genes |
| Strain, strain background (*P. aeruginosa*) | PAO1 Rg Δ*psl* | *Rybtke et al., 2012* | | In frame deletions of *wspF*, *pslBCD* genes |
| Strain, strain background (*P. aeruginosa*) | PAO1 Rg Δ*cdrA* | *Rybtke et al., 2015* | | In frame deletions of *wspF*, *cdrA* genes |
| Cell line (*Homo sapiens*) | Caco-2 | ATCC | HTB-37 RRID:CVCL_0025 | |
| Cell line (*Canis*) | MDCK | Sigma Aldrich | 84121903-1VL RRID:CVCL_0422 | |
| Cell line (*Mus musculus*) | CMT-93 | ATCC | RRID:CCL-223 RRID:CVCL_1986 | |
| Recombinant DNA reagent | pFY_113 (plasmid) | *Berk et al., 2012* | | Plasmid for generation of in-frame *rbmA* deletion mutants |
| Recombinant DNA reagent | pFY_330 (plasmid) | *Berk et al., 2012* | | Plasmid for generation of in-frame *bap1* deletion mutants |
| Recombinant DNA reagent | pFY_922 (plasmid) | *Fong et al., 2010* | | Plasmid for generation of in-frame *vpsL* deletion mutants |
| Recombinant DNA reagent | pNUT1236 (plasmid) | *Hartmann et al., 2019* | | Arabinose-inducible plasmid containing the coding region of *rbmA* |
| Recombinant DNA reagent | pBAD/Myc-His B (plasmid) | *Fong and Yildiz, 2007* | | Arabinose-inducible plasmid containing the coding region of *bap1* |
| Chemical compound, drug | Lithium phenyl-2,4,6-trimethylbenzoyl phosphinate (LAP) | Tokio Chemical Industries | | |
| Chemical compound, drug | PEGDA (MW 3400, 6000, 10000) | Biochempeg | | |
| Chemical compound, drug | PEGDA (MW 700) | Sigma-Aldrich | | |
| Software | Fiji | Fiji | | |
| Software | OriginPro | OriginLab Corporation | | |
| Software | MATLAB | Mathworks | | |
| Software | Imaris | Bitplane | | |
| Algorithm | 3D TFM | *Toyjanova et al., 2014* | | |
| Other | SYTO9 stain | Invitrogen | S34854 | 10 μM |
| Other | CellTracker Orange CMRA stain | Invitrogen | C34551 | 10 μM |
| Other | Hoechst stain | Thermo Fischer Scientific | 62249 | 5 μg/ml |
| Other | Calcein-AM stain | Sigma Aldrich | 17783 | 5 μM |

## Cell culture

CMT-93, Caco-2, and MDCK cells were maintained in T25 tissue culture flasks (Falcon) with DMEM medium (Gibco) supplemented with 10% fetal bovine serum at 37°C in a $CO_2$ incubator. Cell lines are frequently screened for mycoplasma and were authenticated by STR profiling.

## Cell culture on collagen/Matrigel gels

To resemble the extracellular matrix natural niche, we cultured epithelial cells at the surface of collagen and Matrigel-based hydrogels. Hydrogel solutions were prepared on ice to avoid premature gelation by mixing 750 µl of neutralized collagen with 250 µl of growth-factor reduced Matrigel matrix (Corning, 356231). The neutralized collagen was obtained by mixing 800 µl of native type I collagen isolated from the bovine dermis (5 mg/ml, Cosmo Bio Co., Ltd.) with 10 µl of $NaHCO_3$ (1 M), 100 µl of DMEM-FBS and 100 µl of DMEM 10X. We then spread 100 µl of the hydrogel solution in glass bottom dishes (P35G-1.5–20 C, MatTek), which were kept on ice. Excess solution was removed from the sides of the well to avoid the formation of a meniscus. To promote collagen adhesion, the wells were previously functionalized with a 2% polyethyleneimine solution (Sigma-Aldrich) for 10 min and a 0.4% glutaraldehyde solution (Electron Microscopy Science) for 30 min. We finally placed the coated dishes at 37°C in a $CO_2$ incubator for 20 min to allow gelation.

MDCK, CMT-93, and Caco-2 cells were detached from the flask using trypsin (Sigma-Aldrich). We seeded the cells at a concentration of 1000 cells/$mm^2$ on top of the gels. We let the cells adhere for 1 day and then we filled the dishes with 2 ml of culture medium. The medium was changed every 2 days.

## Bacterial strains and culture conditions

A list of the strains and plasmids is provided in the Key Resources Table. All strains were grown in LB medium at 37°C. Only Vc Rg ΔrbmA pBADrbmA and Vc Rg Δbap1 pBADbap1, were grown in LB containing 0.5 wt% arabinose and respectively gentamicin and ampicilin before inoculation in the microfluidic channels.

Deletion of the *V. cholerae* genes *rbmA,bap1 and vpsL* were generated by mating a parental A1552 *V. cholerae* strain, rugose variant, with *E. coli S17* strains harboring the deletion constructs according to previously published protocols (*Fong et al., 2006*). The *rbmA* complementation strain was generated by tri-parental mating using *E. coli S17* harboring an arabinose-inducible *rbmA* gene and a helper strain. The *bap1* complementation strain was generated by electroporation of an arabinose inducible plasmid containing the coding region of *bap1* inside the deletion mutant.

*P. aeruginosa* strains (PAO1 parental strain) are all constitutively expressing GFP (attTn7::miniTn7T2.1-Gm-GW::PA1/04/03::GFP).

## Infection of tissue-engineered epithelia by *Vibrio cholerae*

*V. cholerae* was grown in LB medium at 37°C to mid-exponential phase (OD 0.3–0.6). Bacteria were washed three times by centrifugation and resuspension in Dulbecco's phosphate-buffered saline (D-PBS). The cultures were then diluted to an optical density of $10^{-7}$ and filtered (5.00 µm-pore size filters, Millex) to ensure the removal of large bacterial clumps, thereby isolating planktonic cells. This ensured that biofilms growing on epithelia formed from single cells. We loaded 200 µL of diluted culture on top of mammalian cells that were cultured for 1 to 7 days post-confluence on collagen/Matrigel gels. Bacteria were allowed to adhere to the surface for 20 min, after which cells were rinsed two times with D-PBS.

Biofilm were grown under flow after seeding of Vc Rg on top of CMT-93 cells, while they were grown in stationary conditions for MDCK and Caco-2 cells. For the implementation of the flow on top of CMT-93 cells, we prepared a circular slab of PDMS with the same dimensions as the dish. We punched 1 mm inlet and outlet ports in this PDMS slab. We then glued it to the rim of the dish, where no cells are present. We then connected the inlet port to a disposable syringe (BD Plastipak) filled with culture medium using a 1.09 mm outer diameter polyethylene tube (Instech) and a 27G blunt needle (Instech). The syringes were mounted onto a syringe pump (KD Scientific) positioned inside a $CO_2$ incubator at 37°C. The volume flow rate was set to 50 µL·$min^{-1}$. For stationary biofilm growth on top of MDCK cells, the glass bottom dishes were filled with 2 mL of culture medium and were incubated at 37°C in a $CO_2$ incubator.

## Fabrication of PEG hydrogels and mechanical characterization

To generate PEG hydrogels films we prepared solutions of M9 minimal medium containing poly(ethylene glycol) diacrylate (PEGDA) as the precursor and lithium phenyl-2,4,6- trimethylbenzoylphosphinate (LAP, Tokio Chemical Industries) as the photoinitiator. Molecular weight and concentration of

PEGDA were tuned to obtain hydrogels with different stiffnesses (Table 1), while the concentration of LAP is kept constant at 2 mM.

To incorporate fluorescent microparticles into the PEG hydrogels, we modified the original solution by substituting 2 µL (for a solution with a final volume of 100 µL) of M9 medium with 2 µL of red fluorescent particles solution (ThermoFischer, FluoSpheres, Carboxylate-modified Microspheres, 0.1 µm diameter, 2% solids, F8887).

To prepare the samples for mechanical characterization, we filled PDMS wells (5 mm diameter, 4 mm height) with the hydrogel solution. We covered the wells with a coverslip and we let them polymerize in a UV transilluminator (Bio-Rad Universal Hood II) for 5 min. The resulting hydrogel cylinders were immersed in M9 overnight and tested with a rheometer (TA instruments) in compression mode, at a deformation rate of 10 µm/s. Beforehand, the diameter of the cylinders was measured with a digital caliper, while the height of the cylinder was defined as the gap distance at which the force starts differing from zero. The elastic modulus corresponds to the slope of the linear fit of the stress-strain curves in the range of 15% strain. The final modulus is the average modulus of three replicates.

## Fabrication of thin PEG hydrogel layers and implementation with PDMS microfluidic chip

We fabricated microfluidic chips following standard soft lithography techniques. More specifically, we designed 2 cm-long, 2 mm-wide channels in Autodesk AutoCAD and printed them on a soft plastic photomask. We then coated silicon wafers with photoresist (SU8 2150, Microchem), with a thickness of 350 µm. The wafer was exposed to UV light through the mask and developed in PGMEA (Sigma-Aldrich) in order to produce a mold. PDMS (Sylgard 184, Dow Corning) was subsequently casted on the mold and cured at 70°C overnight. After cutting out the chips, we punched 1 mm inlet and outlet ports. We finally punched a 3 mm hole right downstream of the inlet port. This hole, after being covered with a PDMS piece, acts as a bubble trap.

To obtain thin and flat hydrogel layers, a drop of about 80 µL of the hydrogel solution was sandwiched between two coverslips and incubated in the UV transilluminator for 5 min to allow gelation. The bottom coverslip (25 × 60 mm Menzel Gläser) was cleaned with isopropanol and MilliQ water, while the upper one (22 × 40 mm Marienfeld) was functionalized with 3-(Trimethoxysilyl)propyl methacrylate (Sigma-Aldrich) following the standard procedure. In short, cleaned coverslips were immersed in a 200 mL solution of ethanol containing 1 mL of the reagent and 6 ml of dilute acetic acid (1:10 glacial acetic acid:water) for 5 min. They were subsequently rinsed in ethanol and dried. This functionalization enables the covalent linkage of the hydrogel to the coverslip.

Right after polymerization, the coverslips were separated using a scalpel and thus exposing the hydrogel film surface. We then positioned the PDMS microfluidic chip on top of the hydrogel film. This results in a reversible, but sufficiently strong bond between the hydrogel and the PDMS, allowing us to use the chips under flow without leakage for several days. The assembled chips were filled with M9 to maintain the hydrogel hydrated.

## Biofilm growth in microfluidic chambers

All *V. cholerae* and *P. aeruginosa* strains were grown in LB medium, unless specified, at 37°C until mid-exponential phase (OD 0.3–0.6). The cultures were diluted to an optical density of $10^{-3}$ and subsequently filtered (5.00 µm-pore size filters, Millex) to ensure the removal of large bacterial clumps. We then loaded 6.5 µL of the diluted bacterial culture in the channels, from the outlet port. We let

**Table 1.** Molecular weight and concentrations of the precursors used for the generation of the hydrogels and resulting elastic modulus.

| Precursor | Concentration wt/vol | Modulus kPa |
|---|---|---|
| PEGDA MW 10000 (Biochempeg) | 10% | 12.1 ± 0.8 |
| PEGDA MW 6000 (Biochempeg) | 10% | 38.3 ± 1.0 |
| PEGDA MW 3400 (Biochempeg) | 10% | 30.9 ± 2.0 |
| PEGDA MW 700 (Sigma-Aldrich) | 15% | 203.3 ± 13.7 |

them adhere for 20 min before starting the flow. We connected the inlet port to a disposable syringe (BD Plastipak) filled with the medium and mounted onto a syringe pump (KD Scientific), using a 1.09 mm outer diameter polyethylene tube (Instech) and a 27G needle (Instech). The volume flow rate was 10 μL·min$^{-1}$, which corresponds to a mean flow speed of about 0.25 mm·s$^{-1}$ inside the channels. The biofilms were grown at 25°C in LB, unless specified. For Vc Rg ΔrbmA pBADrbmA, both liquid cultures and biofilms were grown in LB containing 30 μg/ml of gentamicin and 0.5 wt% arabinose. For Vc Rg Δbap1 pBADbap1, both liquid cultures and biofilms were grown in LB containing 100 μg/ml of ampicilin and 0.5 wt% arabinose. For *V. cholerae* N16961 and C6706 we loaded the bacterial cultures at an optical density of 0.5 and we let them adhere for 1 hr before starting the flow at a volume flow rate of 2 μL·min$^{-1}$. Unless specified the medium used was LB. For some conditions, we used M9 minimal medium supplemented with 2 mM MgSO4, 100 μM CaCl2, 0.5% glucose, MEM Vitamins, and 15 mM triethanolamine (pH 7.1).

## Staining procedures

Mammalian cells were incubated for 20 min in a 10 μM solution of CellTracker Orange CMRA (Invitrogen, C34551) and washed with DPBS before seeding the bacteria. For standard visualization of biofilm grown on top of epithelial monolayers, since *V. cholerae* strains were not constitutively fluorescent, samples were incubated for 20 min with a 10 μM solution of SYTO9 (Invitrogen, S34854) and washed with DPBS before visualization. This results in double staining of epithelial cells. For the visualization of live and dead cells in infected monolayers we instead incubated the samples for 20 min in a solution containing 5 μg/ml Hoechst (Thermo Fischer Scientific, 62249) and 5 μM Calcein-AM (Sigma Aldrich, 17783). For the visualization of epithelial cells monolayers permeability, we added 1 ml of a 2 μM solution of fluorescein isothiocyanate-dextran (Sigma Aldrich, 46944) on top of the cells and imaged after 30 min.

*V. cholerae* biofilms grown in microfluidic channels were incubated for 20 min with a 10 μM solution of SYTO9 (Invitrogen, S34854) and washed with M9 minimal medium before visualization.

## Visualization

For all visualizations, we used an Nikon Eclipse Ti2-E inverted microscope coupled with a Yokogawa CSU W2 confocal spinning disk unit and equipped with a Prime 95B sCMOS camera (Photometrics). For low magnification images, we used a 20x water immersion objective with N.A. of 0.95, while for all the others we used a 60x water immersion objective with a N.A. of 1.20. We used Imaris (Bitplane) for three-dimensional rendering of z-stack pictures and Fiji for the display of all the other images.

To obtain the deformation profiles, z-stacks of the hydrogel containing fluorescent microparticles were performed every 0.5 μm, while a brightfield image of the base of the biofilm was taken to allow measurement of the diameter of the biofilm. For the visualization of the full biofilm, z-stacks of the samples were taken every 2–3 μm. For timelapse experiments, biofilms were imaged as soon as the flow was started, while for all the other experiments biofilms were imaged between 10 and 48 hr post-seeding.

## Image analysis and computation of deformation profiles

Starting from confocal imaging pictures of the microparticle-containing hydrogel, we aimed at identifying the gel surface and extracting quantitative information about its deformation induced by the biofilms. In most cases, we used an automated data analysis pipeline as described below. To get an average profile of the deformation caused by the biofilms, we performed a radial reslice in Fiji over 180 degrees around the center of the deformation (one degree per slice). We then performed an average intensity projection of the obtained stack. To calculate the diameter of the biofilm, we averaged 4 measurements of the biofilm diameter taken at different angles. The resliced images were then imported in Matlab R2017a (Mathworks) as two-dimensional (x-y) matrices of intensities. In these images, the surface was consistently brighter than the rest of the gel. Therefore, we identified the surface profile as the pixels having the maximal intensity in each column of the matrix. Note that the bottom of the gel sometimes also comprised bright pixels that introduced noise in the profile. To reduce this problem, we thus excluded 20 rows at the bottom of each image (~3.7 μm). We then calculated the baseline position of our gel – namely, the height of the non-deformed portion of the

gel. In our pictures, this corresponds to the height at the left and right extremities of the profile. Therefore, we defined the baseline as the average of the first 50 and last 50 pixels of the profile (~9 µm on each side of the profile). We then offset the whole picture so that the baseline position corresponded to y = 0. We undersampled the extracted surface profiles to further reduce noise, by keeping only the maximal y value over windows of 40 pixels. Finally, we fitted a smoothing spline to the undersampled profile using the built-in *fit* function in Matlab, with a smoothing parameter value of 0.9999.

To quantify the deformation that biofilms induced on the hydrogel, we measured the amplitude ($\delta_{max}$) of the deformed peak and its full width at half maximum ($\lambda$). First, we evaluated the fitted profile described above at a range of points spanning the whole width of the picture and spaced by 0.0005 µm. We identified the maximal value of the profile at these points, which corresponds to the amplitude of the peak $\delta_{max}$ (with respect to the baseline, which is defined as y = 0). We then split the profile in two: one part on the left of the maximum, and one part on its right. On each side, we found the point on the profile whose y value was the closest to $0.5 \cdot \delta_{max}$ using the Matlab function *knnsearch*. We then calculated the distance between their respective x values, which corresponds to the $\lambda$ of the deformed peak. Our data analysis program also included a quality control feature, which prompted the user to accept or reject the computed parameters. When imaging quality was insufficient to ensure proper quantification with our automated pipeline, we measured the deformation manually in Fiji. Graphs were plotted with OriginPro.

## Digital volume correlation and traction force microscopy

We performed particle tracking to measure local deformations and ultimately compute stress and traction forces within hydrogels as biofilms grew. To do this, we performed timelapse visualizations of the hydrogel during the formation of a biofilm at high spatial resolution with a 60X, NA 0.95 water immersion objective. We thus generated 200 µm x 200 µm x 25 µm (50 stacks of 1200 $\times$ 1200 pixels) volumes at 14 different time points. These images were subsequently registered to eliminate drift using the Correct 3D Drift function in Fiji. To compute local material deformations which we anticipated to generate large strains, we used an iterative Digital Volume Correlation (DVC) scheme (*Toyjanova et al., 2014*). These were performed with 128 $\times$ 128$\times$64 voxel size in cumulative mode, meaning deformations are calculated by iterations between each time point over the whole 4D timelapse, rather than directly from the reference initial image. The DVC code computes material deformation fields in 3D which we subsequently use as input for the associated large deformation traction force microscopy (TFM) algorithm (*Toyjanova et al., 2014*). The TFM calculates stress and strain fields given the material's Young modulus ($E$ = 38 kPa in our case) and Poisson ratio (0.459 taken from measurements for polyacrylamide [*Takigawa et al., 1996*]) to ultimately generate a traction force map at the hydrogel surface.

## Acknowledgements

We thank Fitnat Yildiz, Matt Parsek, Melanie Blokesch, Bonnie Bassler and Knut Drescher for strains and plasmids, and Carey Nadell, John Kolinski and Pedro Reis for discussions.

## Additional information

### Funding

| Funder | Author |
| --- | --- |
| Swiss National Science Foundation | Alice Cont<br>Tamara Rossy<br>Zainebe Al-Mayyah<br>Alexandre Persat |
| Cavaglieri Foundation | Alice Cont<br>Tamara Rossy<br>Zainebe Al-Mayyah<br>Alexandre Persat |
| Fondation Beytout | Alice Cont<br>Tamara Rossy |

|  | Zainebe Al-Mayyah<br>Alexandre Persat |
|---|---|
| Gebert Rüf Stiftung | Alice Cont<br>Tamara Rossy<br>Zainebe Al-Mayyah<br>Alexandre Persat |

The funders had no role in study design, data collection and interpretation, or the decision to submit the work for publication.

## Author contributions

Alice Cont, Conceptualization, Data curation, Formal analysis, Investigation, Methodology, Writing - original draft; Tamara Rossy, Data curation, Formal analysis, Methodology, Writing - original draft; Zainebe Al-Mayyah, Resources, Investigation; Alexandre Persat, Conceptualization, Supervision, Funding acquisition, Writing - original draft, Project administration

## Author ORCIDs

Alice Cont (ID) https://orcid.org/0000-0001-6224-3743
Alexandre Persat (ID) https://orcid.org/0000-0001-8426-8255

## Decision letter and Author response

Decision letter https://doi.org/10.7554/eLife.56533.sa1
Author response https://doi.org/10.7554/eLife.56533.sa2

# Additional files

## Supplementary files

- Transparent reporting form

## Data availability

Source data files have been provided for all figures, tables and figure supplements.

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
