## [Decision Letter]

**Acceptance summary:**

Your data describing how biofilms generate force sufficient to deform surface epithelia are important and add nuance to our understanding of the role of biofilms in pathogenesis. The addition of data on wild-type cells is particularly valuable, as it reinforces the original findings and supports the argument that surface deformation is a physiologically relevant consequence of biofilm formation in the host.

**Decision letter after peer review:**

Thank you for submitting your article "Biofilms deform soft surfaces and disrupt epithelia" for consideration by *eLife*. Your article has been reviewed by Gisela Storz as the Senior Editor, a Reviewing Editor, and three reviewers. The following individuals involved in review of your submission have agreed to reveal their identity: Howard A Stone (Reviewer #1).

The reviewers have discussed the reviews with one another and the Reviewing Editor has drafted this decision to help you prepare a revised submission.

Summary:

All three reviewers were impressed by the quality of the experiments and the accompanying analysis. As a group, they felt that data describing how biofilms generate force sufficient to deform surface epithelia are important and add to our understanding of the role of biofilms in pathogenesis. At the same time, there was concern that the use of *Vibrio cholerae* and *Pseudomonasaeruginosa* strains that overproduce ECM rather than wild type strains did not directly address the main claim being made in the paper and so reduced the study's potential impact. All three reviewers are in agreement that additional experiments assessing the impact of wild type *V. cholerae* and *P. aeruginosa* strains on the deformation of soft surfaces are warranted. Copies of the original reviewer comments are included at the end of this letter. While requests for additional experiments are limited to those described above with wild type bacterial strains, all three reviewers had suggestions to improve the clarify and quality of the manuscript that we hope you will find useful.

Reviewer #1:

The authors study how biofilms (*V. cholerae* and *P. aeruginosa* and their distinct EPS compositions) grow on and deform soft substrates (mostly soft synthetic hydrogels), with excellent visualization of the three-dimensional deformations and quantitative measurements of the surface stresses using traction force microscopy. They then show that the associated stresses, in the MPa range, can disrupt epithelial monolayers, which is a model system consistent with physiological materials. I am familiar with some of the (mostly recent) literature on deformation associated with biofilms on soft substrates, but I am not aware of any controlled studies of biofilms growing and deforming soft, tissue-like materials consisting of mammalian cells, as the authors do here, where they document the magnitude of the stress and the disruption of soft epithelial cell monolayers (enterocyte-like Caco-2 cells). They then highlight this as a potential means for a mechanical mode of infection. I think the work is very good and very well illustrated. Whereas some of the results and conclusions overlap thematically with recent studies on biofilms deforming soft substrates (e.g. Yan et al., 2019), it goes beyond existing studies with very informative 3D imaging of the deformed shapes, stress measurements, and mechanical damage of the underlying soft materials, and as noted above offers the first evidence that the stresses associated with biofilms growth can disrupt epithelial monolayers, which I do not believe has been recognized, or documented, as a mechanical route to damage of tissues. I think the paper provides valuable new ideas, conclusions, and implications. I did not have any substantial concerns. Admittedly, the most important original conclusion on demonstrating disruption of model epithelial monolayers would be enhanced if some more direct connection was possible to even qualitative observations in clinical medicine or tissue engineering.

Reviewer #2:

This is an interesting manuscript that investigates the consequences of mechanical force generated by biofilm communities on its surrounding environment and the intrinsic factors that contribute to generating mechanical forces by these communities. They use two of the paradigm species for studying biofilms in the laboratory: *Pseudomonasaeruginosa* and *Vibrio cholerae*. They find that these species generate significant mechanical force that can perturb and deform the surrounding environment. They do so in a biofilm matrix-dependent manner and these forces have the potential to influence a eukaryotic host, suggesting that they may contribute to pathogenesis.

Overall, I really liked this manuscript. I think the findings are important and novel. Some of the methodologies are quite creative. Perhaps the biggest weakness of this manuscript is that some of the important points are really well developed and could benefit from additional experimentation (specific suggestions are listed below).

Specific comments:

1) Figure 2A- The schematic is a bit confusing- It looked like a bacterial aggregate to me at first- I think I would provide the silhouette of a bacterial aggregate above it.

2) The role of matrix components in generating mechanical forces could be more developed. For *V. cholerae*, RbmC and Vps mutants could be tested. Do matrix component overexpression impact deformation? The same with *P. aeruginosa*- does matrix overexpression have expected effects?

3) Other surface relevant appendages influence deformation strength/kinetics- MSHA pili for Vc or type IV pili for Pa? Does rhamnolipid surfactant influence deformation?

4) Mutant strains should be complemented.

5) A key set of experiments is depicted in Figure 5. The data are promising, but fairly qualitative. Characteristics of the monolayer can be measured- such as membrane integrity. What happens in terms of eukaryotic cell survival?

Reviewer #3:

In most cases, the mechanisms by which biofilms impact bacterial pathogenesis remain to be elucidated. Research into biofilm-associated pathogenesis has largely focused on either; the coordination of virulence factor and/or toxin production associated with biofilms, or biofilm-associated invasion of the host immune response. Here Cont et al., artfully demonstrate that simple mechanical forces resulting from the process of biofilm formation are strong enough to deform soft hydrogel surfaces and invoke damage to host epithelium. This work is particularly insightful given the established importance of biofilms to the lifecycles and pathogenesis of the two model organisms studied. This work in highly impactful, and offers significant new insights into the impact of biofilms on bacterial infection and pathogenesis.

One major point of consideration, is that the authors focus on *Vibrio cholerae* and *Pseudomonasaeruginosa* strains that over produce biofilm matrix components. What would be the effect of strains that aren't EPS over producers? Would they generate similar forces to EPS over producers, or lower forces that are still enough force to disrupt soft surfaces and/or epithelial cells? Or is part of the force generated due to the overabundance of matrix components? Perhaps further analysis of impacts of a wildtype strain on deformation of hydrogel surfaces would enhance the impact of this work, and further demonstrate the impact of these findings.

For analysis of *V. cholerae* and *P. aeruginosa* impacts on soft surfaces in vitro, the experimental setup utilizes a very low flow rate of 10μL/min. The flowrate of fluid through the small intestine where *V. cholerae* establishes infection can vary widely, but averages 2.5mL/min in fasting individuals, and can exceed 20mL/min in individuals after meals [Fine et al., 1995]. Can the authors comment on how increasing the flow rate impact the forces exerted by the biofilms on soft surfaces?

The authors utilize *Vibrio cholerae* interactions with Caco-2 as a model for biofilm interactions with host epithelial cells. These experiments nicely show that the forces generated by *V. cholerae* biofilms can interfere with epithelial cell-cell interactions to disrupt their integrity. *V. cholerae* largely colonizes portions of the small intestines, where cells are covered by a thick layer of mucin which at physiologic pH can range in stiffness from 11-163 Pa [Sotres et al., 2017]. Given that this is much below the stiffness of the weakest hydrogel tested (12 kPa), can the authors comment on the ability of *V. cholerae* to form biofilms on such a weak substrate. Would it serve to provide access to the underlying epithelial layers?

References:

Fine KD, Santa Ana CA, Porter JL, Fordtran JS. 1995; Effect of changing intestinal flow rate on a measurement of intestinal permeability. Gastroenterology. 108: 983-989. doi:10.1016/0016-5085(95)90193-0

Sotres J, Jankovskaja S, Wannerberger K, Arnebrant T. 2017; Ex-Vivo Force Spectroscopy of Intestinal Mucosa Reveals the Mechanical Properties of Mucus Blankets. Sci Rep. 7: 1-14. doi:10.1038/s41598-017-07552-7

---

## [Author Response]

Summary:All three reviewers were impressed by the quality of the experiments and the accompanying analysis. As a group, they felt that data describing how biofilms generate force sufficient to deform surface epithelia are important and add to our understanding of the role of biofilms in pathogenesis. At the same time, there was concern that the use of *Vibrio cholerae* and *Pseudomonas aeruginosa* strains that overproduce ECM rather than wild type strains did not directly address the main claim being made in the paper and so reduced the study's potential impact. All three reviewers are in agreement that additional experiments assessing the impact of wild type *V. cholerae* and *P. aeruginosa* strains on the deformation of soft surfaces are warranted. Copies of the original reviewer comments are included at the end of this letter. While requests for additional experiments are limited to those described above with wild type bacterial strains, all three reviewers had suggestions to improve the clarify and quality of the manuscript that we hope you will find useful.

We thank the editor and the reviewers for their constructive comments which we think greatly improved the quality of our manuscript. We provide the responses to the specific comments below. However, we would like to first comment here on the main comment on the effect of EPS overproduction and its comparison with WT strains.

To address this critical point, we performed extensive experimental characterizations of the differences in substrate deformation behaviors between WT and rugose *V. cholerae* and *P. aeruginosa*. These results are now shown in two new figures: Figure 3 and Figure 3—figure supplement 1. The first important conclusion of these additional experiments is that WT strains are also able to deform soft substrates. Second, we observed quantitative differences between WT and rugose variants which we attribute to their distinct abilities to secrete EPS. More specifically, in *P. aeruginosa*, WT formed smaller biofilms which deformed the substrate to a lesser extent than the rugose mutant (Figure 3A). In *V. cholerae*, WT cells had the ability to deform the substrate when grown in media conditions allowing for biofilms to form robustly. WT could deform the substrate as much as the rugose, but with more heterogeneity between biofilms (Figure 3C). We went even further as to test the deformation ability of multiple *V. cholerae* strain backgrounds which have known differences in their ability to form biofilms. These strains showed distinct ability to deform hydrogels. In particular, N16961 and C6706 deform the substrate more dramatically than A1552 (Figure 3—figure supplement 1). Altogether, these experiments allow us to demonstrate that deformations are not an artefact from matrix hyper-secretion and is therefore relevant to WT biofilms. The additional experiments also help us better comprehend the contributions of matrix production in the onset of deformations. In addition to the new figures, we modified the manuscript text to discuss these results in subsection “Wild-type and rugose biofilms deform soft-substrates” and the Discussion section.

Reviewer #2:This is an interesting manuscript that investigates the consequences of mechanical force generated by biofilm communities on its surrounding environment and the intrinsic factors that contribute to generating mechanical forces by these communities. They use two of the paradigm species for studying biofilms in the laboratory: *Pseudomonas aeruginosa* and *Vibrio cholerae*. They find that these species generate significant mechanical force that can perturb and deform the surrounding environment. They do so in a biofilm matrix-dependent manner and these forces have the potential to influence a eukaryotic host, suggesting that they may contribute to pathogenesis.Overall, I really liked this manuscript. I think the findings are important and novel. Some of the methodologies are quite creative. Perhaps the biggest weakness of this manuscript is that some of the important points are really well developed and could benefit from additional experimentation (specific suggestions are listed below).Specific comments:(1) Figure 2A- The schematic is a bit confusing- It looked like a bacterial aggregate to me at first- I think I would provide the silhouette of a bacterial aggregate above it.

We thank the reviewer for this comment. We clarified Figure 2A by adding dashed lines delineating the biofilm edge as in Figure 2B.

(2) The role of matrix components in generating mechanical forces could be more developed. For *V. cholerae*, RbmC and Vps mutants could be tested. Do matrix component overexpression impact deformation? The same with P. aeruginosa- does matrix overexpression have expected effects?

We thank the reviewer for this comment. We performed additional experiments to address this point. First, we performed a detailed investigation of the effect of matrix expression levels on the magnitude of substrate deformations. We refer to the response to the Summary for more information. In summary, we show that WT strains have the potential to deform the substrates as much as overproducing strains. Second, we tested a *vpsL* mutant lacking the ability to secret Vps polysaccharide. Consistent with previous reports, this mutant could not form biofilms and we could not observe any substrate deformation as now shown in Figure 4—figure supplement 1A. Our attempts at generating a *rbmC* deletion mutant were not fruitful for reasons we could not control. We however predict that it would produce a similar phenotype as the *bap1* deletion mutant given the similarity of their function in substrate adhesion.

3) Other surface relevant appendages influence deformation strength/kinetics- MSHA pili for Vc or type IV pili for Pa? Does rhamnolipid surfactant influence deformation?

We agree with the reviewer that pili might play a role in substrate attachment. This is true for single cells in the initial steps of surface colonization. However, little is known about the function of pili within a biofilm in subsequent stages of its formation. Still, we visualized deformation generated by the strain PAO1 *ΔpilA*. We observed that its biofilms can still deform the substrate but to a lower extent compared to PAO1 WT. We show the results below. We however decided to not include this result in the main text as we cannot interpret this data precisely. In fact, the contribution of type IV pili in biofilm mechanics remains undefined as do rhamnolipids and are future research directions for us and others. Similarly, we anticipate that the MSHA pili are only involved in initial adhesion events and do not contribute to biofilm adhesion at later stages (J. Teschler et al., 2015).

4) Mutant strains should be complemented.

We agree with the reviewer for pointing this out. We complemented the mutant that showed clear phenotypes. Thus, we complemented *V. cholerae* Rg *ΔrbmA* and *Δbap1*. We show the results in Figure 4—figure supplement 1B-C.

5) A key set of experiments is depicted in Figure 5. The data are promising, but fairly qualitative. Characteristics of the monolayer can be measured- such as membrane integrity. What happens in terms of eukaryotic cell survival?

We thank the reviewer for her/his/their feedback and agree that further characterization of the tissue would be informative to understand the impact of biofilms on the epithelium. As a result, we performed complementary experiments that further demonstrate the impact of disruptions on the monolayer. We focused on measuring epithelial cell viability and epithelial permeability. We performed viability measurements with the fluorescent indicator Calcein-AM. These visualizations showed that epithelial cells in the biofilms were dead, but cells remaining in the monolayer stayed alive. We performed permeability measurement using a fluorescent Dextran assay. These visualizations show a dramatic increase in permeability upon disruption by *V. cholerae* biofilms. We now include these results in Figure 6—figure supplement 1 and Figure 6—figure supplement 2 and discuss them in the main text in subsection “Biofilms deform and disrupt epithelial cell monolayers” and the Discussion section.

Reviewer #3:In most cases, the mechanisms by which biofilms impact bacterial pathogenesis remain to be elucidated. Research into biofilm-associated pathogenesis has largely focused on either; the coordination of virulence factor and/or toxin production associated with biofilms, or biofilm-associated invasion of the host immune response. Here Cont et al., artfully demonstrate that simple mechanical forces resulting from the process of biofilm formation are strong enough to deform soft hydrogel surfaces and invoke damage to host epithelium. This work is particularly insightful given the established importance of biofilms to the lifecycles and pathogenesis of the two model organisms studied. This work in highly impactful, and offers significant new insights into the impact of biofilms on bacterial infection and pathogenesis.One major point of consideration, is that the authors focus on *Vibrio cholerae* and *Pseudomonas aeruginosa* strains that over produce biofilm matrix components. What would be the effect of strains that aren't EPS over producers? Would they generate similar forces to EPS over producers, or lower forces that are still enough force to disrupt soft surfaces and/or epithelial cells? Or is part of the force generated due to the overabundance of matrix components? Perhaps further analysis of impacts of a wildtype strain on deformation of hydrogel surfaces would enhance the impact of this work, and further demonstrate the impact of these findings.

We thank the reviewer for the feedback and suggestions which we think make our conclusions stronger. We commented on this point in the Summary. In summary, our new results demonstrate that WT cells can also deform soft substrates, and that the amplitude of deformation likely depends on the amount of matrix production. In particular, we observe that WT has the ability to deform soft substrates as much as the rugose strain, demonstrating their similar abilities to generate force. We however notice that the deformations are less consistent between WT biofilms. The new results are in Figure 3 and Figure 3—figure supplement 1 and in the text in subsection “Wild-type and rugose biofilms deform soft-substrates” and the Discussion section.

For analysis of *V. cholerae* and *P. aeruginosa* impacts on soft surfaces in vitro, the experimental setup utilizes a very low flow rate of 10μL/min. The flowrate of fluid through the small intestine where *V. cholerae* establishes infection can vary widely, but averages 2.5mL/min in fasting individuals, and can exceed 20mL/min in individuals after meals [1]. Can the authors comment on how increasing the flow rate impact the forces exerted by the biofilms on soft surfaces?

We thank the reviewer for the comment. We would like to clarify that the biofilm experiences forces generated by flow near the wall. The magnitude of the flow rate across the entire section of the fluidic system does not capture this shear force. To demonstrate this, we can calculate the shear stress at the surface of our microchannel and at the intestinal epithelium (for a discussion of flow forces on biofilms, see Persat et al., 2015 and Dufrene and Persat, 2020). The flow rates yield shear stress values of maximum 10^-2^ Pa in the microchannel and 10^-4^ in the intestine. These stresses are at least seven orders of magnitude smaller than the ones we are considering in this work (100 kPa). They therefore generate negligible forces acting on the biofilm.

The authors utilize *Vibrio cholerae* interactions with Caco-2 as a model for biofilm interactions with host epithelial cells. These experiments nicely show that the forces generated by *V. cholerae* biofilms can interfere with epithelial cell-cell interactions to disrupt their integrity. *V. cholerae* largely colonizes portions of the small intestines, where cells are covered by a thick layer of mucin which at physiologic pH can range in stiffness from 11-163 Pa [2]. Given that this is much below the stiffness of the weakest hydrogel tested (12 kPa), can the authors comment on the ability of *V. cholerae* to form biofilms on such a weak substrate. Would it serve to provide access to the underlying epithelial layers?

We thank the reviewer for raising this point. In fact, we are convinced that the mucus layer is an important aspect that we were unfortunately not able to incorporate in our study. This is mainly due to the current lack of experimental cellular systems producing mucus. However, we predict that biofilms would even more strongly deform the mucus layer due to their very low elasticity. However, the mucus hydrogel is a viscoelastic material which is highly structured and heterogeneous with pore sizes above 500 nm. Previous explorations of *V. cholerae*-mucus interactions suggest that single bacteria can easily penetrate the mucus layer by swimming (Teschler et al., 2015 and Bartlett et al., 2017). Therefore, single cells could not attach on the mucus surface but would rapidly reach the epithelium. To specifically address this point, we modified the manuscript in the Discussion section.